# Focusing Characteristics and Widefield Imaging Performance of the Silicon Metalens in the Visible Range

**DOI:** 10.3390/mi13081183

**Published:** 2022-07-27

**Authors:** Miao Zhao, Binbin Yu, Jing Du, Jing Wen

**Affiliations:** 1Laboratory of Micro-Nano Optoelectronic Materials and Devices, Shanghai Institute of Optics and Fine Mechanics, Chinese Academy of Sciences, Shanghai 201800, China; miaozhao@siom.ac.cn; 2Center of Materials Science and Optoelectronics Engineering, University of Chinese Academy of Sciences, Beijing 100049, China; 3Wenzhou Institute, University of Chinese Academy of Sciences, Wenzhou 325000, China; 4Oujiang Laboratory, Wenzhou 325000, China; 5The Mars Laboratory, Huitong School & Studios, Shenzhen 518052, China; dujing_bio@163.com; 6Engineering Research Center of Optical Instrument and Systems, Ministry of Education and Shanghai Key Lab of Modern Optical System, University of Shanghai for Science and Technology, Shanghai 200093, China

**Keywords:** metalens, imaging performance, focusing characteristics, silicon metalens

## Abstract

Conventional optical high numerical aperture lenses are essential for high-resolution imaging, but bulky and expensive. In comparison, metalens-based optical components are the subjects of intensive investigation for their flexible manipulation of light. Methods of detecting and characterizing focal spots and scanning imaging produced by metalenses are well established. However, widefield imaging by metalenses is experimentally challenging. This study demonstrates the design and realization of silicon-based metalenses with numerical apertures of 0.447 and 0.204 in the broadband spectrum of 580–780 nm for microscopic widefield imaging. The optimized aspect ratio of the single nanorod is 5.1:1, which reduces the fabrication difficulty compared to other, more complicated designs and fabrication. Furthermore, we successfully demonstrate widefield imaging by the designed metalens and compare the simulated and the experimentally extracted modulation transfer function curves of the metalens.

## 1. Introduction

High numerical aperture (NA) lenses play a crucial role in high-resolution imaging, and traditional optical lenses are usually bulky and expensive. Metalenses, however, can realize miniaturization of traditional refractive optical devices into planar structures [1,2,3]. Contrary to traditional optical lenses, metalens-based optical constituents are compact, and can manipulate the phase, amplitude, and polarization [4] of electromagnetic waves at will. These advantages allow metalenses to show strong potential in various applications, such as nonlinear dynamics [5], beam shaping [6,7], high-dimensional holographic images [8,9,10,11], polarization control [4,12,13,14], and analysis [15,16]. In addition to its compact size, the metalens has the advantage of the integration function [17], which can realize specific characteristics, such as multi-spectrum (infrared and microwave) compatibility [18], dual-function switching between holography and display [19], nano-optics imaging [20], off-axis focusing and off-axis dispersion control [21,22], spin-selected and spin-dependent light manipulation [23,24], as well as nonlinear optical chirality, nonlinear geometric Berry phase, and wavefront engineering [25,26].

Commercial lenses are generally prepared by assembling the high-precision lenses accurately, which makes them bulky and expensive. This restricts their application scenarios, as they cannot be integrated into compact and cost-efficient systems. Single-layer planar lenses in the visible range have a particularly high market demand due to their extensive applications in imaging, microscopy, and spectroscopy. Although planar lenses in the visible range can be realized by diffractive components, their composition is based on wavelength-scale structures, so it is difficult to obtain accurate phase profiles and, as a result, it is difficult to achieve high imaging quality with these lenses. Although traditional curved refractive lenses can achieve high imaging quality, they require multiple lenses with complex designs, which must be precisely assembled. They also have the drawbacks of complex designs, bulky systems, and high processing costs, along with a difficult installation setup.

To overcome the aforementioned limitations, this article demonstrates the design and realization of a silicon-based metalens with a low aspect ratio of 5.1:1. The metalens is optimized with numerical apertures of 0.447 and 0.204 for the wavelength of 580–780 nm for widefield microscopic imaging. The calculated modulation transfer function (MTF) curve also illustrates that the resolution of the system is diffraction-limited, i.e., of 1150 lp/mm and 523 lp/mm at the design wavelength of 640 nm, which validates the high imaging quality of the metalens. At NA = 0.204, our MTF results are comparable to Xu, Beibei, et al. [27]; when the MTF is 0.4, its spatial characteristic frequency is approximately 300 lp/mm and when the NA is 0.407, the MTF, i.e., 448 lp/mm, surpasses that of Xu, Beibei, et al. [27]. The results illustrate that in cases of high NA, the MTF of the metalens designed in this article exceeds results reported in the previous literature.

## 2. Materials and Methods

Refractive optical systems rely on gradual phase accumulation by the propagation of light, while diffractive optical systems use amplitude masks or phase masks to transmit light by interference phenomena. As an ultra-thin optical element, the metalens is designed to modify the optical response of the interface by means of its dense arrangement of sub-wavelength resonators. The resonant properties of the scatterer introduce a sudden phase shift in the incident wavefront, making it possible to alter the wavefront of transmitted light at will and realize a new class of planar photonic devices. Our phase manipulation is based on the Pancharatnam-Berry Phase (PB Phase) method [28,29]. By changing the rotation angle of the anisotropic nano-antenna, the circularly polarized incident light is converted into its inverse circularly polarized transmitted light with a geometric phase shift that is twice the rotation angle, in order to realize the above-mentioned phase profile [30,31]. The phase distribution of the spherical focusing beam of the metalens is:(1)$$\varphi \left(r,\lambda \right)=\frac{2\pi }{\lambda }\left(\sqrt{r^{2}+f^{2}}-f\right)$$
where *r* is the radial coordinate in the metalens, and *f* is the focal length at the design wavelength λ. The metalens is designed for a design wavelength of 640 nm and is functional and optimized for 580–780 nm. The unit structure parameters are optimized as shown in Figure 1a, and the refractive index of silicon is shown in Figure 1b. Taking the center of the metalens as the origin, a rectangular coordinate system is established, and silicon is used as the optical material for the unit cell. The height of the unit cell is *H* = 380 nm, which is placed on the quartz substrate. The rotation angle of the unit cell is determined by Equation (1).

All the simulations were performed by using the commercial finite-difference time-domain (FDTD) method implemented by commercial software ‘Lumerical FDTD Solutions’ (produced by Ansys Inc., Vancouver, BC, Canada) [32] to optimize the parameters of the unit cell so that it has a high polarization conversion efficiency in the working wavelength range and can converge an increased number of light beams to achieve a higher focusing efficiency. The optimized unit cell parameters are: period *S* = 275 nm, length *L* = 75 nm, and the width of the unit cell is *W* = 95 nm. The polarization conversion efficiency of the unit cell reaches 92%, as shown by the black curve in Figure 1c. Although the polarization conversion efficiency of the unit cell varies widely, the transmittance of the overall metalens array always remains above 50%, as shown by the red curve in Figure 1c. This is because the coupling effect between the unit cells still exists, which can allow the unit array to exhibit high transmittance, even when the corresponding polarization conversion efficiency is very low. Figure 1d,f depicts the schematic of the metalens.

The whole metalens is then simulated by using the optimized unit structure parameters. Due to the limitation of simulation resources, a structure with a diameter of 10 μm and a focal length of 8 μm was chosen. The simulation results are presented in Figure 1e,g. It can be seen that the metalens can achieve a bright and highly concentrated focal spot and also achieve diffraction-limited focusing at its design wavelength.

## 3. Fabrication and Characterization of the Metalens

After obtaining the ideal simulation results, the next step is to process the metasurface samples. Here, the electron beam etching method is used as a standard approach of lithography to process the samples.

To initiate the sample fabrication, a 380 nm thick amorphous silicon layer is sputtered onto a quartz substrate. The sample is then spin-coated by a 350 nm thick layer of ZEP520 with post-bake at 180° for 1 min. Afterward, electron beam lithography is conducted by JEOL JBX6300 fs operating at an acceleration voltage of 100 keV. Next, the sample is developed in an amyl acetate solution for 65 s at room temperature, succeeded by the inductive coupled plasma-reactive ion etching process. The microscope image of the fabricated metalens is shown in Figure 2a.

To further verify the broadband operation and dispersion characteristics of the metalens, it is necessary to image the focal spots characteristic of the metalens. The optical path system includes the following components: white light laser (NKT Photonics SuperK EXTREME EXR-15), polarizer (Thorlabs, Newton, NJ, USA, LPNIR100-MP), a quarter-wave plate (Thorlabs AQWP10M-1600), metalens, 100× microscope objective lens, tube lens (Thorlabs TL200-3 P), CMOS, and high-precision motorized linear stage (Thorlabs MLS230-1). The schematic of the experimental setup is shown in Figure 2b.

In this experiment, the metalens functions as a transmissive structure, and a microscope system needs to be built behind the device to observe the generated focal spot. As opposed to a traditional optical lens, the metalens can impose phase distribution with a resolution of subwavelength dimensions. The focusing of the metalens is nearly diffraction-limited since the focusing is spherical aberration-free under the field of view angle of 0°. First, collimate the output light of the super-continuum white light laser and then pass the beam through the polarizer and quarter-wave plate, respectively. The axial direction of the quarter-wave plate is 45° to the polarization direction of the polarizer in order to generate circularly polarized light. The circularly polarized light is then incident perpendicular to the transmissive metalens and converges into a focal spot on the other side of the device.

When the laser beam is strictly coaxial, first place the 100× high numerical aperture objective lens on the motorized linear stage close to the side of the metalens, so that the high numerical aperture objective lens can clearly image the metalens structure on the substrate and record the position of *L*1. When the position of the motorized linear stage is *L*1, move the stage in order to move the high numerical aperture objective lens away from the metalens. When the light passing through the high numerical aperture objective lens forms a circular spot with the minimum size and the strongest intensity on the CMOS, the position *L*2 of the motorized linear stage is recorded and the calculation of *L*2-*L*1 is the focal length *f* of the metalens. Then, change the wavelength of the incident light and repeat the above steps to obtain the focal length of the metalens at different wavelengths. The focal spot of the metalens under different numerical apertures is shown in Figure 3.

## 4. Imaging Performance of the Metalens

After experimentally measuring the focal spot of the metalens, imaging analysis of the metalens is further carried out. The experimental setup is shown in Figure 4a. The setup includes the following components: micro projector, polarizer, quarter waveplate, tube lens, 50× objective, metalens, 30× objective, quarter waveplate, polarizer, and CCD.

In this experiment, a micro projector, a polarizer, a quarter-wave plate, a tube lens, and a 50× objective lens are used to produce the “object” needed for imaging; the polarizer and quarter-wave plate are used to modulate the imaging beam to circularly polarized light. A tube lens and a 50× objective lens are used to reduce the image projected by the projector to the field of view of the metalens, while the metalens is used for imaging, and the 30× objective lens is used to collect the imaging beam of the metalens. A quarter-wave plate and a polarizer are used to filter the circularly polarized light in its original polarization state. During the experiment, metalenses with different numerical apertures are used for imaging, as shown in Figure 4(b1–b3), which depicts the objects projected by the projector, while Figure 4(b4–b6) renders the images produced by the metalens with NA = 0.447; and Figure 4(b7–b9) renders the images produced by the metalens with NA = 0.204. It can be seen via Figure 4b that, despite the image color distortion and reduced sharpness caused by uneven efficiency and chromatic aberration, the metalens has adequate imaging quality. Comparing the second and third rows of Figure 4b, the imaging quality with higher NA of 0.447 is better than that with a lower NA of 0.204. However, the distortion with higher NA is larger. The uneven efficiency also causes the imaging color to shift to the high-efficiency wavelength. The silicon-based metalens designed in this paper has a far greater efficiency in red light than blue-violet light, resulting in distortion of the image color and the overall image taking a reddish tone. Secondly, the extremely strong “negative dispersion” characteristic of the metalens causes the sharpness of the picture to decrease. Correcting the chromatic aberration and optimizing the uniformity of the spectral efficiency will be an important direction for future work on the optimal design of the metalens. The field of view of the metalens is FOV=2*atan(IH/EFL), where IH is the image height and EFL is the focal length. When NA is 0.204 and 0.447, the corresponding values of FOV are 25° and 45°. Differing from the work of A. Martins [33], we adopt a broadband design of the metalens based on the PB phase, which can achieve wide-field imaging in a wide wavelength range. In addition, due to its unique optimization and design, the metalens can be easily processed by an electron beam etching method, with a relatively low aspect ratio of the single nano-rod unit (5.1:1), a fact which is not considered in Z. Huang’s article [34].

Reference is made to the limit resolution formula of the microscope objective: *δ* = 0.61λ/NA. It can be seen that the theoretical resolution of the metalens is 0.87 μm and 1.91 μm, and the characteristic frequencies are 1 mm/3.15 μm = 523 lp/mm and 1 mm/0.87 μm = 1150 lp/mm. In order to quantify the influence of different NA on the imaging quality of the metalens, we calculate the corresponding modulation transfer function (MTF) curve using Lumerical FDTD Solutions. Additionally, to save the computational resources, we simulate the metalens with a diameter *D* of 10 μm, NA of 0.204, and 0.447, and a design wavelength of 640 nm.

Figure 5 shows the MTF curves of the metalens with NA of 0.204 and 0.447 for various incident wavelengths. The modulation transfer function comprehensively reflects the imaging quality of the designed metalenses. It can be seen from Figure 5 that the first nodes which exhibit zero value for the simulated and experimental MTFs coincide as to place. For the case of NA = 0.204, the simulated and experimental resolutions are 448 lp/mm and 391 lp/mm, respectively, at the design wavelength of 640 nm when the value of MTF is 0.4. For the case of NA = 0.447, the simulated and experimental resolutions are 640 lp/mm and 303 lp/mm, respectively. In contrast, with the increase of NA, the MTF will increase with a similar characteristic frequency, which implies that with the increase of NA, the imaging quality of the metalens is enhanced. For NA of 0.447, however, the MTF of the simulation and the MTF of the experiment is quite different, which is caused by errors in the imaging process and the inevitable aberrations in the fabrication. Despite this variance, the experimental MTF envelope is close to the simulated MTF curve.

## 5. Conclusions

This article proposes a design for a metalens with different numerical apertures based on the geometric phase method. First, based on the wavefront manipulation, the phase distribution formula of the planar metalens is deduced. The numerical simulation software FDTD is then used to model and simulate the designed metalens and the structural parameters of the metalens are optimized to achieve the highest possible conversion efficiency in the working bandwidth, i.e., 580–780 nm. An experiment is then carried out to test the light field distribution and the focusing characteristics of the metalens with different numerical apertures, and analyze its focusing characteristics. Afterward, the imaging performance of the metalens with different numerical apertures is analyzed experimentally. The experimentally extracted modulation transfer function (MTF) curves comprehensively reflect that the resolution of the system has reached 391 lp/mm and 448 lp/mm for NA = 0.204 and 0.447 at the wavelength of 640 nm.

The theoretical and experimental results obtained establish that the metalens has the potential for microscopic widefield imaging and that it can be utilized in various optical systems to achieve high integration and miniaturization of the systems. Meanwhile, these results validate the idea that uneven efficiency and dispersion have become important factors limiting the imaging quality of the metalens, which has a definite significance for the optimization of the design of the metalens in the future.

## Figures and Tables

**Figure 1 micromachines-13-01183-f001:**
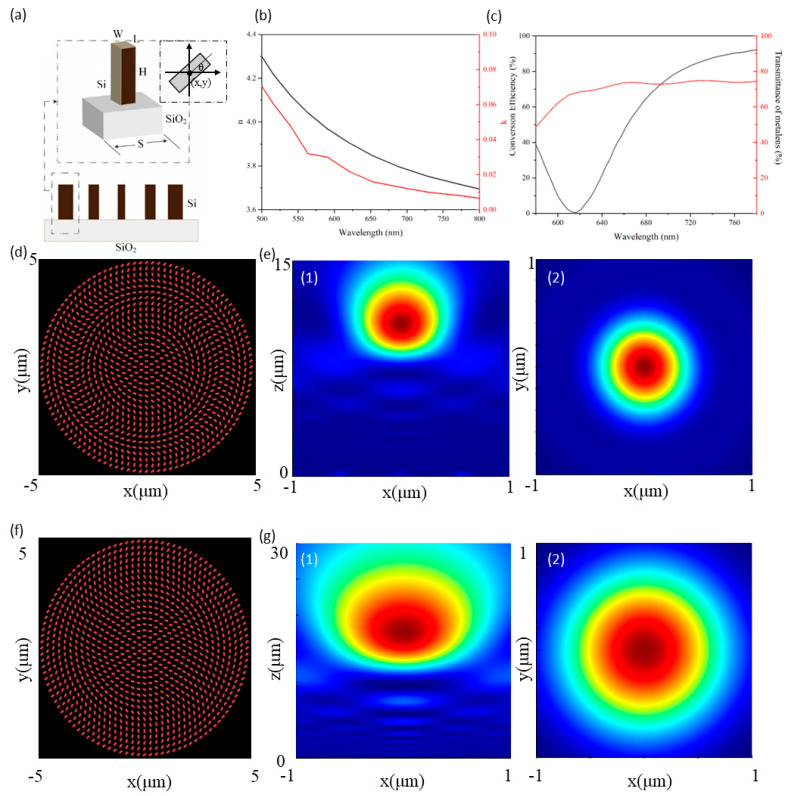
(**a**) Schematic diagram of the rotation angle and structure of the nano-antenna; (**b**) the refractive index of silicon, where the black curve is n and the red curve is k; (**c**) polarization conversion efficiency of the unit cell (black curve) and transmittance of metalens (red curve); (**d**) schematic diagram of the metalens; (**e**) simulation results of the metalens at NA = 0.447: (**1**) normalized intensity distribution of the metalens at the x-z plane, (**2**) normalized intensity distribution of the metalens at the x-y plane; (**f**) schematic diagram of the metalens; (**g**) simulation results of the metalens at NA = 0.204: (**1**) normalized intensity distribution of the metalens at the x-z plane, (**2**) normalized intensity distribution of the metalens at the x-y plane. The design wavelength is 640 nm. The colormap represents the normalized distribution of light field intensity, from blue to red, as the distribution of light intensity.

**Figure 2 micromachines-13-01183-f002:**
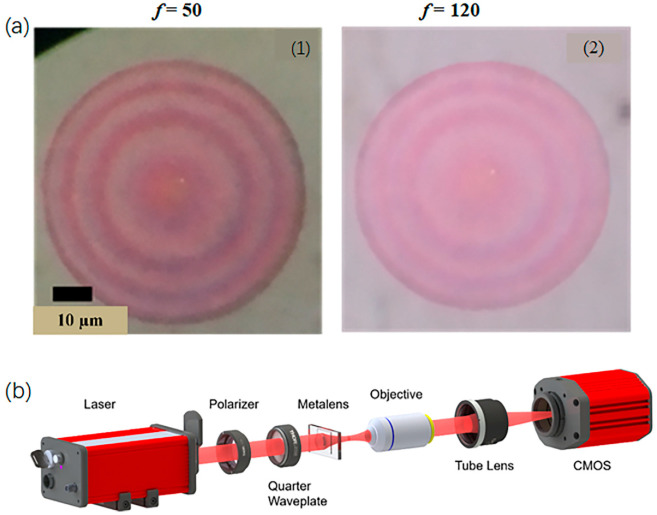
(**a**) Images of the fabricated metalens under the optical microscope (**1**) *f* = 50 µm and NA = 0.447 (**2**) *f* = 120 µm and NA = 0.204; (**b**) schematic for the optical setup used for characterizing the focusing characteristics of the designed metalens.

**Figure 3 micromachines-13-01183-f003:**
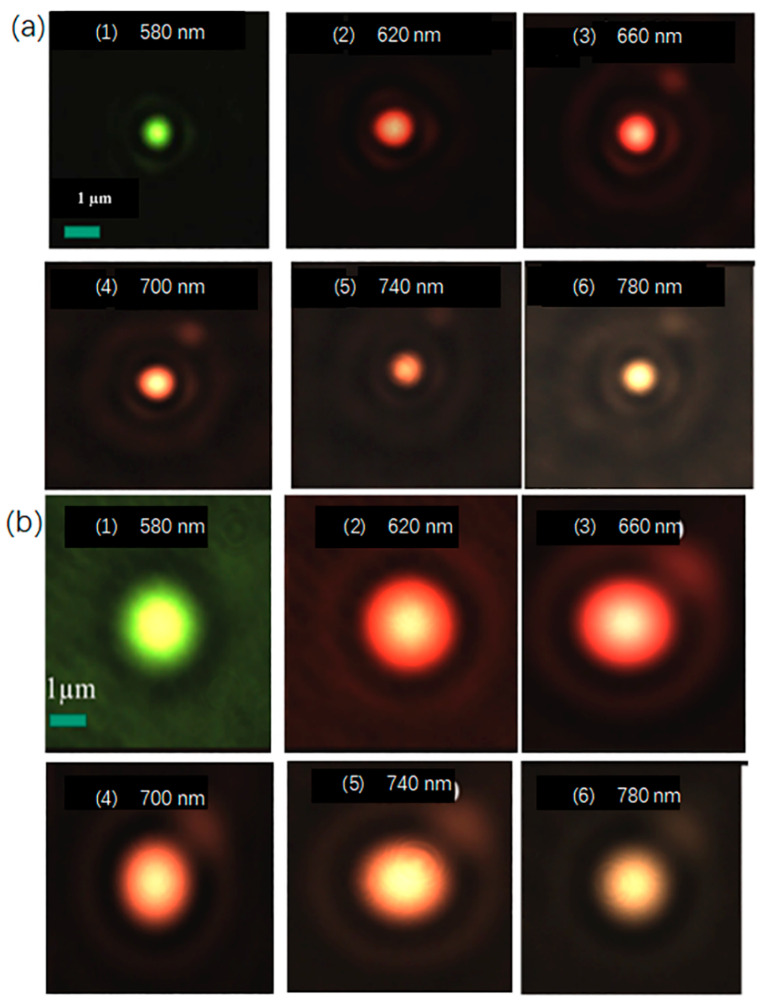
The experimental intensity distribution of the metalens along focal planes at various incident wavelengths with different NA (Numerical Aperture): (**a**) with focal length = 50 µm, NA = 0.447; and (**b**) with focal length = 120 µm, NA = 0.204.

**Figure 4 micromachines-13-01183-f004:**
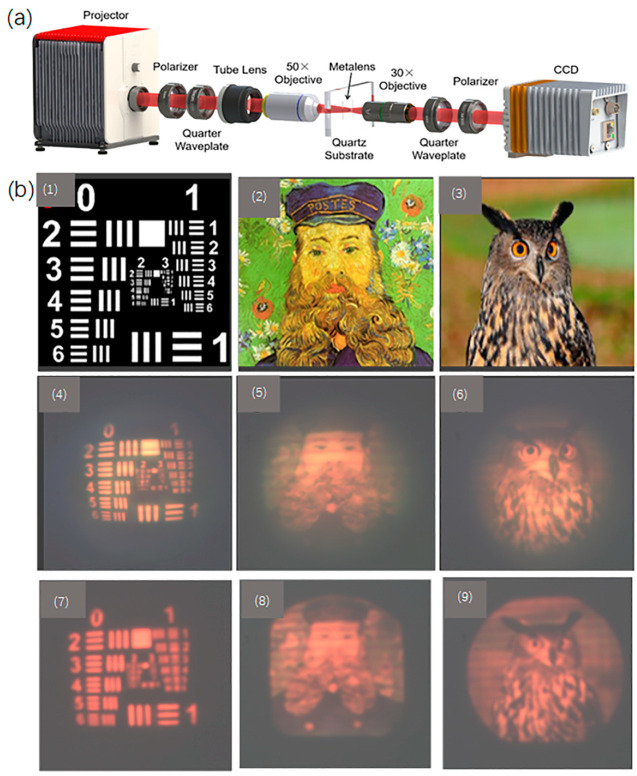
(**a**) The optical setup for characterizing the imaging performance of the designed metalens; (**b**) imaging performance of the metalens. (**1**)–(**3**) are the source images, (**4**)–(**6**) are the images produced by the designed metalens with NA = 0.447, while (**7**)–(**9**) are the images produced by the designed metalens with NA = 0.204. (**1**) 1951 USAF Resolution Target; (**2**) Vincent Van Gogh’s *Portrait of the Postman Joseph Rulan*; (**3**) gran búho, pájaro photo by Ton W, an animal image from pixnio.com (accessed on 24 June 2022) in November 2018.

**Figure 5 micromachines-13-01183-f005:**
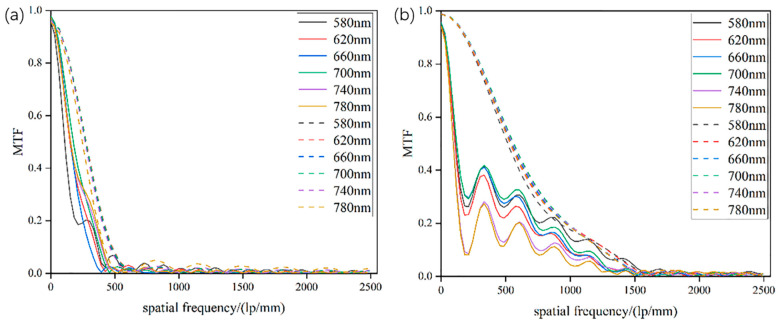
The MTF curve of the focal point on the focal plane of the metalens with normal incident NA of (**a**) 0.204 and (**b**) 0.447 at different incident wavelengths. Dashed lines are simulated MTF, while solid lines are extracted from the experimental results of Figure 3.

## Data Availability

The data that support the findings of this study are available from the corresponding author upon reasonable request.

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
