# Peer review of "Focusing Characteristics and Widefield Imaging Performance of the Silicon Metalens in the Visible Range"

_micromachines, 2022, doi:10.3390/mi13081183_

Round 1
Reviewer 1 Report
This study describes the design and development of a silicon-based metalens with the ability to operate at the visible spectrum for widefield microscopic imaging. Extensive analyses and assessments were carried out to illustrate the performance of the system in practical imaging tasks. The authors also provide quantitative results from the NA and other important parameters of the devised system. The work contains interesting results and has an immense potential to be published in this journal. The work is in a good shape; however, the authors must improve the quality of the Introduction by considering the role of meta-optics or nonlinear metaphotonics in the emergence of metalenses. Useful information in that respect can be found in: Nature Reviews Materials 2, 17010 (2017), Materials Today 51, 208-221 (2021).
Author Response
Response letter
We appreciate very much for your constructive comments on our manuscript. We have revised our manuscript according to your comments. The detailed revisions are summarized as follows.
Comments from Reviewer 1 and our responses
Comment 1
- This study describes the design and development of a silicon-based metalens with the ability to operate at the visible spectrum for widefield microscopic imaging. Extensive analyses and assessments were carried out to illustrate the performance of the system in practical imaging tasks. The authors also provide quantitative results from the NA and other important parameters of the devised system. The work contains interesting results and has an immense potential to be published in this journal. The work is in a good shape; however, the authors must improve the quality of the Introduction by considering the role of meta-optics or nonlinear metaphotonics in the emergence of metalenses. Useful information in that respect can be found in: Nature Reviews Materials 2, 17010 (2017), Materials Today 51, 208-221 (2021).
Response 1
The comment is important.
The newly added references are shown as follows:
- Li, G.; Zhang, S.; Zentgraf, T. Nonlinear photonic metasurfaces. Nature Reviews Materials 2017, 2, 5, doi:10.1038/natrevmats.2017.10.
- Ahmadivand, A.; Gerislioglu, B. Deep-and vacuum-ultraviolet metaphotonic light sources. Materials Today 2021, 51, doi:10.1016/j.mattod.2021.05.019.
- Page 1, Starting from line 42
“as well as nonlinear optical chirality, nonlinear geometric Berry phase and wavefront engineering [25,26].” is added in the manuscript.

Reviewer 2 Report
This paper presents a metalens that performs widefield imaging with a high numerical aperture. Silicon nanorods have been used to create the metalens. What is missing in this paper is how is this different from what other people have already achieved. For example, (https://opg.optica.org/oe/fulltext.cfm?uri=oe-29-9-13542 and https://pubs.acs.org/doi/full/10.1021/acsphotonics.0c00479?)
The paper fails to highlight the innovation compared with such other papers, it is therefore not clear what is the main contribution of this paper to the literature.
So I think the authors need to re-think their research design and better explain the significance of their work compared with other similar works.
Author Response
Response letter
We appreciate very much for your constructive comments on our manuscript. We have revised our manuscript according to your comments. The detailed revisions are summarized as follows.
Comments from Reviewer 2 and our responses
Comment 1
- This paper presents a metalens that performs widefield imaging with a high numerical aperture. Silicon nanorods have been used to create the metalens. What is missing in this paper is how is this different from what other people have already achieved. For example, (https://opg.optica.org/oe/fulltext.cfm?uri=oe-29-9-13542 and https://pubs.acs.org/doi/full/10.1021/acsphotonics.0c00479?)
The paper fails to highlight the innovation compared with such other papers, it is therefore not clear what is the main contribution of this paper to the literature.
So I think the authors need to re-think their research design and better explain the significance of their work compared with other similar works.
Response 1
The comment is important. We consider our work to be different from the reference mentioned by the reviewer. Their work is also excellent. The difference between our work and the reference (titled with “On Metalenses with Arbitrarily Wide Field of View”) is that we use a geometric phase that works over a broad wavelength range for imaging, while the reference uses a propagation phase that works in a single wavelength.
The difference between our work and the other reference (titled with “Achromatic and wide-field metalens in the visible region”) is that our work uses parameters that can be quickly optimized and easily fabricated. While a double-layer metasurface is difficult to fabricate. And this design did not discuss how to align the double layers in fabrication process.
The newly added references are shown as follows:
- Martins, A.; Li, K.; Li, J.; Liang, H.; Conteduca, D.; Borges, B.; Krauss, T.; Martins, E. On metalenses with arbitrarily wide field of view. ACS Photonics 2020, 7, 8. doi:10.1021/acsphotonics.0c00479.
- Huang, Z.; Qin, M.; Guo, X.; Yang, C.; Li, S. Achromatic and wide-field metalens in the visible region. Optics Express 2021, 29, 9, doi:10.1364/OE.422126.
* Page 7, Starting from line 190
Compared to the first reference, we also calculated FOV in our experiments. We add the following comments: “The field of view of the metalens is FOV=2*atan(IH/EFL), where IH is the image height and EFL is the focal length. When NA is 0.204 and 0.447, the corresponding values of FOV are 25° and 45°. Different from the work of Martins A [33], we adopt a broadband design of the metalens based on the PB phase, which can achieve wide-field imaging in a wide wavelength range. In addition, due to the unique optimization and design, the metalens can be easily processed by electron beam etching method with a relatively low aspect ratio of the single nano-rod unit (5.1:1), which is not considered in Huang Z’s article [34].”

Reviewer 3 Report
The authors design and realize silicon-based metalens with a low aspect ratio of 5.1:1. The metalens is optimized with numerical apertures of 0.447 and 0.204 for the wavelength of 580–780 nm for widefield microscopic imaging. This work contains some interesting results. However, the authors need to clarify the following issues before a final recommendation can be made:
1. The complex permittivity of Si should be given.
2. In Fig1.(c), the polarization conversion efficiency of the unit cell is 0 in the wavelength of 610nm,the focusing effect should be described.
3. Why is the diffraction-limited focusing achieved ? Specific explanation should be given in the article.
4. Another work of metalens (show as follow) should be discussed and cited.
(1) Optics Express, 2021, 29(26): 43270-43279.
(2) Nanomaterials, 2019, 9(5): 761.
Author Response
Response letter
We appreciate very much for your constructive comments on our manuscript. We have revised our manuscript according to your comments. The detailed revisions are summarized as follows.
Comments from Reviewer 3 and our responses
Comment 1
- The authors design and realize silicon-based metalens with a low aspect ratio of 5.1:1. The metalens is optimized with numerical apertures of 0.447 and 0.204 for the wavelength of 580–780 nm for widefield microscopic imaging. This work contains some interesting results. However, the authors need to clarify the following issues before a final recommendation can be made:
The complex permittivity of Si should be given.
Response 1
The comment is important. We add the refractive index of Si which is given in Figure 1(b).
- Page 3, Starting from line 108
“(b) The refractive index of silicon;”
Comment 2
In Fig1.(c), the polarization conversion efficiency of the unit cell is 0 in the wavelength of 610nm,the focusing effect should be described.
Response 2
It is a very useful comment. We calculate the transmittance of the metalens array which is updated in Figure 1(c) and add the corresponding explanation in the manuscript.
- Page 3, Starting from line94
“The polarization conversion efficiency of the unit cell is as high as 92%, as shown by the black curve in Figure 1(c). Although the polarization conversion efficiency of the unit cell varies in a large range, the transmittance of the overall metalens array always remains above 50% as shown by the red curve in Figure 1(c). This is because the coupling effect between the unit cells still exists, which can make the unit array have a good transmittance even when the corresponding polarization conversion efficiency is very low.” is added in the manuscript.
Comment 3
Why is the diffraction-limited focusing achieved? Specific explanation should be given in the article.
Response 3
Thanks for reviewer’s comment. We have added an explanation in the manuscript that metalens can achieve diffraction-limited focusing.
- Page 5, Starting from line139
“In this experiment, the metalens is a transmissive structure and a microscope system needs to be built behind the device to observe the generated focal spot. Different from traditional optical lenses, the metalens can impose phase distribution with a resolution of subwavelength dimension. The focusing of the metalens is nearly diffraction-limited since the focusing is spherical aberration-free under the field of view angle of 0°.” is added in the manuscript.
Comment 4
Another work of metalens (show as follow) should be discussed and cited.
(1) Optics Express, 2021, 29(26): 43270-43279.
(2) Nanomaterials, 2019, 9(5): 761.
Response 4
The comment is important.
The newly added references are shown as follows:
- Wang, W.; Yang, Q.; He, S.; Shi, Y.; Liu, X.; Sun, J.; Guo, K.; Wang, L.; Guo Z. Multiplexed multi-focal and multi-dimensional SHE (spin Hall effect) metalens. Optics Express 2021, 29, 26, doi:10.1364/OE.446497.
- Wang, W.; Zhao, Z.; Guo, C.; Guo, K.; Guo, Z. Spin-selected dual-wavelength plasmonic metalenses. Nanomaterials 2019, 9, 5, doi:10.3390/nano9050761.
- Page 1, Figure 41
“spin-selected and spin-dependent light manipulation [23,24]” is added in the manuscript.

Round 2
Reviewer 2 Report
It seems better now. I am fine with it